# Printable and Machinable Dental Restorative Composites for CAD/CAM Application—Comparison of Mechanical Properties, Fractographic, Texture and Fractal Dimension Analysis

**DOI:** 10.3390/ma14174919

**Published:** 2021-08-29

**Authors:** Wojciech Grzebieluch, Piotr Kowalewski, Dominika Grygier, Małgorzata Rutkowska-Gorczyca, Marcin Kozakiewicz, Kamil Jurczyszyn

**Affiliations:** 1Laboratory for Digital Dentistry, Department of Conservative Dentistry Witch Endodontics, Wroclaw Medical University, Krakowska 26, 50-425 Wroclaw, Poland; 2Department of Fundamentals of Machine Design and Mechatronic Systems, Wroclaw University of Science and Technology, Lukasiewicza 7/9, 50-371 Wroclaw, Poland; piotr.kowalewski@pwr.edu.pl; 3Department of Vehicle Engineering, Faculty of Mechanical Engineering, Wroclaw University of Science and Technology, Lukasiewicza 5, 50-371 Wroclaw, Poland; dominika.grygier@pwr.edu.pl (D.G.); malgorzata.rutkowska-gorczyca@pwr.edu.pl (M.R.-G.); 4Department of Maxillofacial Surgery, Medical University of Lodz, 113 S. Zeromski Street, 90-549 Lodz, Poland; marcin.kozakiewicz@umed.wroc.pl; 5Department of Oral Surgery, Wroclaw Medical University, Krakowska 26, 50-425 Wroclaw, Poland; kamil.jurczyszyn@umed.wroc.pl

**Keywords:** fractography, texture analysis, fractal dimension analysis, dental CAD/CAM materials, printable resin composites

## Abstract

Thanks to the continuous development of light-curing resin composites it is now possible to print permanent single-tooth restorations. The purpose of this study was to compare resin composites for milling -Gandio Blocks (GR), Brilliant Crios (CR) and Enamic (EN) with resin composite for 3D printing—Varseo Smile Crown plus (VSC). Three-point bending was used to measure flexural strength (σ*_f_*) and flexural modulus (E*_f_*). The microhardness was measured using a Vickers method, while fractographic, microstructural, texture and fractal dimension (FD) analyses were performed using SEM, optical microscope and picture analysis methods. The values of σ*_f_* ranged from 118.96 (±2.81) MPa for EN to 186.02 (±10.49) MPa for GR, and the values of E*_f_* ranged from 4.37 (±0.8) GPa for VSC to 28.55 (±0.34) GPa for EN. HV01 ranged from 25.8 (±0.7) for VSC to 273.42 (±27.11) for EN. The filler content ranged from 19–24 vol. % for VSC to 70–80 vol. % for GR and EN. The observed fractures are typical for brittle materials. The correlation between FD of materials microstructure and E*_f_* was observed. σ*_f_* of the printed resin depends on layers orientation and is significantly lower than σ*_f_* of GR and CR. E*_f_* of the printed material is significantly lower than E*_f_* of blocks for milling.

## 1. Introduction

Human teeth play an important role in aesthetics, chewing and occlusion. Extensive hard tissue loss caused by caries, root canal treatment, wear or fractures requires, in many cases, indirect restorative procedures to restore function and appearance of the tooth. The constant development of technology resulted in the widespread use of CAD/CAM systems, both laboratory and in-office ones, for the manufacturing of indirect single teeth reconstruction.

The advantages of CAD/CAM technology are based on the simplicity of clinical procedure of indirect dental restorations fabrication at a reduced time and cost. The dominant production method is the subtractive manufacturing of solid materials (blocks and discs) using CNC machines [1]. The portfolio of materials suitable for subtractive manufacturing of permanent restorations covers a wide range of clinical indications including metals, ceramic and resin composites materials [2,3,4].

An alternative to subtractive manufacturing is additive manufacturing, which is the process of depositing layers of material to create a 3D object [5]. The printing process takes place layer by layer, by sintering the powder, depositing a molten thermoplastic material or by light curing of the resin [6]. This manufacturing method is commonly used for the manufacturing of metal frameworks, templates, temporary restorations, splints, and removable prostheses [1,4]. The marginal fit of 3D printed restorations is comparable with the fit of the milled ones [7]. Thanks to the continuous development of light-curing resin composite materials, it is now possible to print adhesively cemented permanent single-tooth restorations [8]. One of the requirements for printable materials is flowable consistency and structural stability during printing and storage. Maintaining a stable liquid consistency suggests that the printable dental material for permanent restorations, similarly to flowable dental composites, should contain less inorganic filler than composites available in blocks and discs. Lower filler content affects the stiffness of the material and may lead to the E-modulus value significantly lower than the one of hard tissues [9,10,11].

According to ISO standards, the recommended procedure for material testing is a 3-point bending test. This method allows the calculation of flexural strength and flexural modulus [12,13]. For a better understanding of the material properties, an additional fractographic analysis is used [14]. An interesting supplement to traditional research methods can be the application of mathematical methods such as texture and fractal dimension analysis.

A fractal dimension (FD) analysis is used in the estimation of complex, irregular shapes or surfaces [15]. In the analysis of complicated shapes, Euclidian geometry may fail. In classic Euclidean geometry, we are used to the fact that the number of dimensions is an integer. For example, a point has zero dimensions, a line has one dimension which is its length, a plane has two dimensions: length and width, and a solid has three dimensions: height, width and length. However, fractals are shapes beyond previously mentioned principles. Their dimensions are not integer and may become values between 0 and 3 dimensions. Fractals can be magnified unlimitedly, and subsequent details of their structure are similar to their initial shape. This feature of fractals is called self-similarity. In daily life, we deal with numerous natural shapes which can be approximately described as fractals, for instance, a network of blood vessels or nerves. There are numerous mathematical methods to calculate the fractal dimension—in our study, the modified box-counting method is used. It enables the analysis of grey scaled images—intensity difference scaling method for assessment of the fractal dimension. FDA is widely used during analyses of positron emission tomography, radiographic images, computed tomography or magnetic resonance images [16,17,18,19].

Texture analysis (TA) is another mathematical method that enables the analysis of the surface. Pixels build a digital image. Every pixel is described by two features: coordinates and colour/brightness. Pixels create a delicate structure of an image named texture. Texture is a collection of recurrent graphical patterns characterized by brightness, entropy, smoothness, uniformity, roughness, granulation, randomness, or linearity [20]. Texture analysis is widely applied in case of magnetic resonance, computed tomography, or X-ray images [21,22,23,24].

This study aimed to compare mechanical properties, fractographic, microstructure, texture and fractal dimension analysis of selected commercially available resin composites used for additive and subtractive manufacturing of permanent single-tooth restorations.

The null hypothesis was that there would be no significant differences in flexural strength, moduli, microhardness, microstructure, fractal dimension and texture analysis between resin composite materials for milling and for printing.

## 2. Materials and Methods

### 2.1. Study Design

The comparison of flexural strength, flexural modulus, microhardness, and fractographic, microstructure, texture and fractal dimension analysis were conducted. The tested composite CAD/CAM materials were Grandio blocs^®^ (VOCO, Cuxhaven, Germany)—GR, Brilliant Crios^®^ (Coltene/Whaledent A.G. Altstatten, Switzerland)—CR, Enamic^®^ (Vita Zahnfabrik, Bad Sackingen, Germany)—EN and VarseoSmile Crown plus^®^ (Bego, Bremen, Germany)—VSC. VarseoSmile Crown plus^®^ is a liquid material for 3D printing. Other materials are delivered in blocks for grinding. Their composition according to the literature is given in Table 1 Machinable materials used in the study. In the case of GR and CR materials, the flexural strength, flexural modulus and microhardness were previously published by the authors [10].

### 2.2. Sample’s Fabrication

The CAD/CAM blocks (GR, CR, EN) were cut with a low-speed water-cooled diamond saw Miracut 151 (Metcon, Bursa, Turkey) to obtain bar-shaped specimens (*n* = 10). Specimens were finished with wet silicon carbide (400 ISO/FEPA, average grain size 35 µm) until dimensions of 15 mm long, 4 mm wide, and 1.5 mm thick (*n* = 10) were reached according to ISO 6872:2015 (accuracy 0.01 mm) [12]. Measurements were performed using a Mitutoyo Digimatic IP65 (MITUTOYO, Kawasaki, Japan) micrometer. The samples were stored dry at room temperature. Sample preparation according to ISO 4049 was not possible due to the size of the material blocks [13].

Two groups of VSC bar samples, 15 mm long, 4 mm wide, and 1.5 mm thick (*n* = 10), were printed using Sonic Mini 4K (Phrozen, Hsinchu City, Taiwan). The group A (VSC A) samples were printed vertically to the platform, the group B (VSC B) samples were rotated in the X and Y axes by 45 degrees (Figure 1). Chitubox 1.8 free software (www.chitubox.com, accessed on 26 July 2021) was used to prepare the G-code for the printer. Resin profile has been optimized for VSC resin, with a layer height 0.05 mm, bottom layer count 8, exposure time 6.5 s, Bottom exposure 20 s, lift distance 5 mm, and lift speed 60 mm/s.

Group A was printed without supports; Group B was printed with automatically generated standard (Chitubox) medium size supports with 75% density. Ethanol was used for post-processing (according to the user manual). Final curing was performed in Form Cure (Formlabs, Somerville, MA, USA), 2 times by 45 min (samples ware rotated after first exposure), with the temperature set on 0. The curing time was set based on microhardness tests (HV01). Microhardness was measured every 5 min until the maximum value was reached. The measurements procedure was repeated 2 times on 5 VSC bars samples. No microhardness increase was recorded after exceeding the specified exposure time (Shimadzu HMV-2T, Shimadzu Corp., Kyoto, Japan).

### 2.3. Mechanical Testing

Flexural properties were measured using a three-point bending test that was conducted with a support span of 12 mm and a speed of 1 mm/min using a universal testing machine LabTest 5.030S LaborTech^®^ (LaborTech Opava, Opava, Czech Republic) equipped with Test&Motion^®^ (LaborTech Opava, Czech Republic) software (in accordance with the ISO 6872:2015) [30]. Prior to the three-point bending test, the width and height of each sample were measured to obtain data for the formulas shown below.

The microhardness was measured by means of a Vickers intender tester (Shimadzu HMV-2T, Shimadzu Corp., Kyoto, Japan) with a load of 980.7 mN (HV 0.1) and dwell time of 10s. Five indentations were applied in a random location for each specimen. Then, the software automatically calculated the hardness value as HV 01. Before the measurement, surfaces of the samples were sequentially polished with composite rubbers HiLusterPlus^®^ Polishing System (Kerr Corp., Orange, CA, USA). Enamic was sequentially polished with a dedicated VITA ENAMIC Polishing Set (Vita Zahnfabrik, Bad Sackingen, Germany). The microhardness measurements were carried out on the same samples that were used in a three-point bending test.

### 2.4. Fractography and Microstructure

Three randomly selected samples of each tested material were used for fractographic and microstructure analysis. The examination was performed using a scanning electron microscope (SEM) Phenom XL (Thermo Fisher Scientific, Waltham, USA) at magnification from 500 up to 10,000× and a stereoptical microscope NIKON AZ 100 (NIKON, Tokyo, Japan) at a magnification of up to 100×. An accelerating voltage of 5–25 kV and SE and BSE detectors were used during the SEM investigation. The samples were cleaned using the ultrasonic cleaner in detergent solution, ethanol and then in deionized water (60 s for each bath). No coating layer was applied to the examined surfaces. The recorded pictures were analysed to identify the fracture mechanism, determine microstructure and the presence of the flaw. Microscopy analysis can provide a report concerning a particle phase size, phase percentage and distribution of phases. Planimetric procedure—Jeffries planimetric method—is based on counting particles in a specified area [31,32]. To perform it, a proper magnification, which provided at least 50 particles, was selected. A circle was drawn on the image, the particles located entirely inside the circle were counted and then the particles intercepting the circle were counted separately and the percentage of the average particles was calculated using the following formula:N_A_ = f(n_inside_ + 0.5n_intercepted_)(1)
where N_A_ is the number of particles per mm^2^ at 1x and f is the Jeffries multiplier.

Jeffries multiplier was calculated according to the formula:f = M2/A(2)
where M is the magnification and A is the area (5000 mm^2^ is the standard size).

### 2.5. Fractal Dimension Analysis

The pictures used for fractography and microstructure analysis were used for fractal dimension and image texture analysis. Two algorithms of fractal dimension counting were used- classical counting box method for 1bit images and intensity difference for 8bit grayscale images. Analysis was performed in ImageJ version 1.53e (Image Processing and Analysis in Java—Wayne Rasband and contributors, National Institutes of Health, USA, public domain license, https://imagej.nih.gov/ij/, accessed on 26 July 2021) and plugin FracLac version 2.5 (Charles Sturt University, Australia, public domain license).

In a classical counting box method of fractal dimension analysis, source images must be a one-bit bitmap (1 for pixel on and 0 for pixel off). Fractal dimension (FD) is calculated using the formula below:(3)$$\mathrm{FD}=\underset{\varepsilon \to 0}{\lim}\frac{\log N\left(\varepsilon \right)}{\log\left(\frac{1}{\varepsilon }\right)}$$
where FD—fractal dimension; ε—length of box side that creates a mesh covering the surface with the examining pattern; *N*(ε)—minimal number of boxes required to cover the examining pattern.

Graphical interpretation of the counting box method is shown in Figure 2.

We used this algorithm for 10,000× magnification images. The 15 µm × 15 µm region of interest (ROI) was cropped from these microstructure images. The range of images is low in the case of such magnification dynamic so the conversion into a 1bit bitmap does not lead to a significant decrease in details.

A modified algorithm of the counting box method which allows the analysis of monochromatic images such as 8 or 16 bits was used. The analysed image is divided into boxes like in the counting box method (Figure 2A). The difference between the maximum and the minimum pixel intensity is counted in each box (δI_i,j,ε_, where i, j—location of the analysed box in a scale ε):

δI_i,j,ε_ = maximum pixel intensity _i,j,ε_—minimum pixel intensity _i,j,ε_.

In the next step, the value of 1 is added to the intensity difference so that the value cannot be 0 (Figure 2B):I_i,j,ε_ = δI_i,j,ε_ + 1(4)

Finally, the fractal dimension of the intensity difference is described by the following formula (Figure 2C):(5)$$\mathrm{FD}\;\mathrm{Idiff}=\underset{\varepsilon \to 0}{\lim}\frac{\ln\left(I\varepsilon \right)}{\ln\left(\frac{1}{\varepsilon }\right)}$$
where FD Idiff—fractal dimension of the intensity difference, Iε = Σ[1δI_i,j,ε_ + 1], ε—scale of box.

All operations are shown in Figure 3.

FD of the intensity difference algorithm was used in the case of the image with 350× magnification. In such magnification, the whole surface of the fracture was visible. Six ROIs at the size 100 µm × 100 µm from each fracture zone (fracture origin, the direction of crack propagation and bending marks on the fracture surface was used for further analysis).

### 2.6. Image Texture Analysis

The surface texture of composite material was evaluated using features derived from two groups (run-length matrix and co-occurrence matrix) and the previously described Texture Index (TI) [33]. The regions of interest (ROIs) were normalised (μ ± 3σ) to share the same average (μ) and standard deviation (σ) of optical density within the ROIs. Selected image texture features (entropy and difference entropy from the co-occurrence matrix a well short- and long-run emphasis moment from the run-length matrix) in ROIs were calculated for each composite material tested:(6)$$Entropy=-\overset{Ng}{\underset{i=1}{\sum }}\overset{Ng}{\underset{j=1}{\sum }}p\left(i,j\right)log(p\left(i,j\right)$$
(7)$$DifEntr=-\overset{Ng}{\underset{i=1}{\sum }}p_{x-y}\left(i\right)log\left(p_{x-y}\left(i\right)\right)$$
where Σ is the sum, Ng is the number of optical density levels in the radiograph, *i* and *j* represent the optical density of pixels that are 5 pixels away from one another, *p* is the probability, and the log is the common logarithm [34],
(8)$$ShrtREmph=\frac{\sum_{i=1}^{Ng}\sum_{k=1}^{Nr}\frac{p\left(i,k\right)}{k^{2}}}{\sum_{i=1}^{Ng}\sum_{k=1}^{Nr}p\left(i,k\right)}$$
(9)$$LngREmph=\frac{\sum_{i=1}^{Ng}\sum_{k=1}^{Nr}k^{2}p\left(i,k\right)}{\sum_{i=1}^{Ng}\sum_{k=1}^{Nr}p\left(i,k\right)}$$
where Σ is the sum, *Nr* is the number of series of pixels with density level *i* and length *k*, *Ng* is the number of levels for image optical density, *Nr* is the number of pixels in series, and *p* is probability [35,36]. Short and long run-length emphasis moment (ShrtREmph, LngREmph) were computed from data taken along the long- axis of the wire, and measures of disarrangement (Entropy and Difference Entropy, i.e., DifEntrp) were computed as non-directional measures. Two of three equations were subsequently used for the Texture Index construction [34]. Finally, the Texture Index (TI), which represents the ratio of the measure of the diversity of the structure observed in the radiograph to the measure of the presence of uniform longitudinal structures, was calculated:(10)$$\begin{array}{ll} Texture\;Index & =\frac{Entropy}{LngREmph}\\ & =\frac{(-\sum_{i=1}^{Ng}\sum_{j=1}^{Ng}p\left(i,j\right)log\left(p\left(i,j\right)\right))\sum_{i=1}^{Ng}\sum_{k=1}^{Nr}p\left(i,k\right)}{\sum_{i=1}^{Ng}\sum_{k=1}^{Nr}k^{2}p\left(i,k\right)} \end{array}$$

And the Composite Index define as:(11)$$\begin{array}{ll} Composite\;Index & =\frac{DifEntrp}{ShrtREmph}\\ & =\frac{-\sum_{i=1}^{Ng}p_{x-y}\left(i\right)log\left(p_{x-y}\left(i\right)\right)\;\sum_{i=1}^{Ng}\sum_{k=1}^{Nr}p\left(i,k\right)}{\sum_{i=1}^{Ng}\sum_{k=1}^{Nr}\frac{p\left(i,k\right)}{k^{2}}} \end{array}$$

### 2.7. Statistical Analysis

Statistica version 13.3 (StatSoft, Cracow, Poland) and Stargraphics Centurion 18 ver.18.1.12 (StarPoint Technologies, Inc., VA, USA) were used to perform all statistical tests. A statistical significance level of 0.05 was assumed. The Shapiro–Wilk test was used to confirm the normality of distribution. As the distribution of samples was normal, parametric statistical tests were performed. The analysis of variance (ANOVA) and the post-hoc least significant difference test were applied to reveal fractal dimension and textural feature differences between the examined microstructures of all materials, the same fracture zones of all materials and between fracture zones of the same material, differences of flexural strength, flexural moduli and microhardness values. The Pearson correlation coefficient (R) was used to estimate the correlation of FD between σ*_f_* and E*_f_*. The correlation coefficient was also calculated to identify relations between filler content by volume and σ*_f_*; filler content by volume and E*_f_*; filler content by volume and HV01.

## 3. Results

### 3.1. Mechanical Testing

Mean flexural strength, flexural modulus and microhardness for all tested materials are shown in Table 2. A statistically significant difference between means of all studied parameters was found.

The values of flexural strength (σ*_f_*) ranged from 118.96 (SD 2.81) MPa for EN to 186.02 (SD 10.49) MPa for GR. The flexural strength of GR was significantly higher in comparison to the other tested materials (*p* < 0.001). Only in pair of EN-VSC A, there were no statistically important differences observed. Flexural strength, in the decreasing order, was as follows: GR > CR > VSC B > VSC A > EN.

The flexural modulus (E*_f_*) values ranged from 4.37 (SD 0.8) GPa for VSC A to 28.55 (SD 0.34) GPa for EN. The flexural modulus of EN was significantly higher in comparison to the other tested materials. Only in pair of VSC A-VSC B, there were no statistically important differences observed. The values of flexural modulus changed in the descending order as follows: EN > GR > CR > VSC B > VSC A.

The values of microhardness ranged from 25.8 (SD 0.7) for VSC A to 273.42 (SD 27.11) for EN. The values of the microhardness of EN were significantly higher compared to other tested materials. Only in pair of VSC A-VSC B, there were no statistically important differences observed. The microhardness values, in the diminishing order, were as follows: EN > GR > CR > VSC B > VSC A.

### 3.2. Fractographic and Structure Analysis

#### 3.2.1. Fractographic Analysis

Observations of the fracture surfaces of all samples revealed the presence of a crack at the outer edge of the sample. The nature of the crack, the presence of the compression curl in the upper part of the fracture surface and the origin in the lower part of the fracture surface indicates bending of the sample (Figure 4, Figure 5, Figure 6, Figure 7 and Figure 8). The use of higher magnifications revealed that new details of the fracture topography, clear delamination and cracks are visible in the material structure. The fracture structure of the CR, GR, VSC A and VSC B samples shows crack deflection due to filler particles bypassing. This process is related to the elongation of the crack path and thus increase in the fracture energy expenditure. Larger and more diversified particle sizes observed in GR and CR samples caused a large deviation in the crack propagation course in relation to VSC A and VSC B. The crack line movement along the interface, after reaching the large particle, undergoes marked deflection and continues propagation almost perpendicular to the original direction. The crack line propagation along the interface, after a large particle reaching, is subject to significant angulation and continues propagation almost perpendicular to the original direction. High magnifications of EN samples revealed the difference in crack propagation—bridging by particles (Figure 6).

#### 3.2.2. Structure Analysis

SEM examination of the polished surfaces of all materials samples revealed the presence of a filler with different particle diameter and differences in the filler content (Table 2 and Figure 9). The filler particle size ranged from 160 nm for CR to 11 µm for EN. The filler volume content ranged from around 19–24 vol. % for VSC B to 70–80 vol. % for GR and EN. The irregular shape of the filler particles was found in all samples. The observation of EN samples revealed bridging between filler particles. This phenomenon was not found in other samples. Pearson’s linear correlation (R) between the filler volume and *σ_f_* was R = 0.39. Strong Pearson’s linear correlation between the filler volume and the E*_f_* R = 0.86 and between filler volume and HV01 R = 0.82 was recorded.

### 3.3. Fractal Dimension Analysis

Mean values of fractal dimension (FD) for 15 µm × 15 µm ROIs are shown in Table 3. The lowest FD is seen in both groups of 3D printed resin composite. It is important to underline that the SD of these two groups is various. SD of the VSC A group is the highest of all groups and approximately two times higher than in the VSC B group. The highest value of FD is observed in the EN group. Results of post-hoc ANOVA statistical test is described in Table 4. There were no statistical differences between FD of VSC A and VSC B group and between CR and GR group. Significant statistical differences are observed between other groups.

Mean values of fractal dimension of fracture zones for 100 µm × 100 µm are shown in Table 5. The lowest value of FD for every fracture site is observed in the EN group. For fracture origin—1.702, the direction of crack propagation (DCP)—1.706 and for bending marks of fracture surface (BM)—1.687.

In all fracture zones, a tendency to reduce FD value was observed. The highest fractal dimension is in the fracture origin, lower in DCP and the lowest in the BM zone. The highest value of FD in fracture origin was seen in the VSC B group (1.780), in DCP and BM site, and the highest value was noted in the VSC B material.

The results of post-hoc (least significant difference) ANOVA between the FD value of the same fracture zone of all examined materials are shown in Table 6. In the fracture origin, we observed statistical differences between all materials expecting VSC A versus VSC B (*p* = 0.302), CR vs. VSC A (*p* = 0.735) and CR vs. VSC B (*p* = 0.175) materials. In the DCP zone, no statistical differences between EN versus GR (*p* = 0.246), VSC A vs. VSC B (*p* = 0.138) were observed. No statistical differences of FD for bending marks on the fracture surface for CR versus EN (*p* = 0.574) and GR (*p* = 0.074), VSC A vs. GR (*p* = 0.435) and VSC B (*p* = 0.335) were revealed.

The results of post-hoc (least significant difference) ANOVA between fractal dimension of fracture origin direction of crack propagation and bending marks on the fracture surface inside the same material are shown in Table 7. We observed a significant difference in fractal dimension value between fracture origin versus bending marks of the fracture surface and DCP versus BM in the VSC A and VSC B group. In the CR group, significant differences between all analysed zones were noted. It is important to emphasize that in EN and GR group there were no significant differences seen between any of the fracture zones. Low linear correlation between FD and σ*_f_* [Mpa] was noted—Pearson’s correlation coefficient R = 0.213. However, a strong linear correlation between FD and E*_f_* [GPa] was observed (R = 0.914).

### 3.4. Image Texture Analysis

The summed Entropy in ROI of VSC materials is quite high and ranks slightly below the CR material (*p* < 0.05) and above the values for EN and GR materials (*p* < 0.05). The interpretation of the surface structure of the studied samples in terms of DifEntrp is not fundamentally different from the analysis based on Entropy (Table 8 and Table 9). The only difference lies in the reversal of the chaos measure values for EN and GR materials. EN, as constructed of the largest inorganic material particles in comparison, has the lowest DifEntrp value (*p* < 0.05). In the VSC-A material, a more random distribution of inorganic particles is observed in both texture features than in VSC-B (*p* < 0.05). It should be emphasized that both investigated ways of calculating ROI entropy indicate the uneven distribution of high entropy places in EN and GR materials, due to the presence of quite large grains in the 15 µm × 15 µm ROI (Figure 10).

The differentiation of the tested materials in terms of surface appearance (*p* < 0.001) was shown in the evaluation of ShrtREmp and LngREmph. It was more pronounced for the short-run emphasis moment, as the EN, VCS A and VCS B materials become more similar in the long-run emphasis moment. The values of these texture features indicated the highest (*p* < 0.05) accumulation of fine image elements (mineral components), while for the GR material that value was the lowest. The VCS A and VSC B materials showed intermediate fragmentation of inorganic phase particles.

Examination of the calculated indices confirmed the differentiation of VSC materials from other dental composites. For Texture Index, the similarity of the surface appearance of VSC A and VSC B composites with a lower value than CR (*p* < 0.05) and a higher value than both EN and GR (*p* < 0.05) can be observed (Table 9). As for the Composite Index, VSC B has a lower value than VSC A (*p* < 0.05) and is similar to CR. On the contrary, VSC A is similar to GR.

Evaluating of fracture origin, it should be noted in the basic data that Entropy was significantly lower (*p* < 0.05) for VSC B than VSC A (in which the texture is among the most chaotically arranged ones) and, in general, it was the lowest among the tested materials. DifEntrp of the surface structure of the fracture origin was similar among the studied materials except for EN (*p* < 0.05), which was a relatively coarse-grained composite with more pronounced surface elements whose presence lowers the value of differential entropy. This site in the frequency analysis of short run-length emphasis moments showed very numerous fine puncta in the VSC A and VSC B materials. They are significantly more healing (*p* < 0.05) than in the other three dental materials. An inverse relationship was shown by the evaluation of long run-length emphasis moments. Extremely few (*p* < 0.05) longitudinal structures at this site in VSC A and VSC B materials could be compared to the other composites tested.

Exceptional homogeneous fine graininess of this site in VSC A and VSC B materials (*p* < 0.05) was similarly indicated by the calculated indices (Table 10, Table 11, Table 12 and Table 13).

Bending marks of investigated materials were much similar than fracture origin and directions of crack propagation, especially for the measures of chaotic textures (Table 14 and Table 15). However, significantly more fine structures (as in the original site in DCP) ware found in VSC A and VSC B materials than in the other three dental composites (*p* < 0.05).

It seems that the differences in the appearance of the fracture surface between the tested materials are best described by the Texture Index. Therefore, the intra-material evaluation of fracture sites is presented with a collective index i.e., Texture Index (Figure 11).

This measure of surface texture increases as the chaotic nature, i.e., randomness and uniformity of the scattering, of texture elements in the ROI increases. It also increases when the number of detected long chains of pixels with similar brightness decreases. For CR, bending marks (*p* < 0.05) have been clearly defined by the fractures, and the origin site and propagation areas are similar. For EN, no differences were found in the three studied regions, although the fracture surface structure varies (low Texture Index values). For GR, the origin location is very homogeneous, in contrast to the propagation areas and bending marks (*p* < 0.05). Both original and propagation areas are very homogeneous in VSC A and slightly more structural heterogeneity can be observed in bending marks locations (*p* < 0.05). VSC B is by far the most homogeneous material in terms of fracture surface appearance and measures of texture features. The origin is not different from propagation area or bending marks.

## 4. Discussion

This paper assessed mechanical properties, fractographic, microstructure, texture and fractal dimension analysis of selected commercially available resin composites used for milling and 3D printing of permanent single-tooth restorations. The null hypothesis was rejected except for flexural strength. No significant difference was found between polymer infiltrated ceramic and printed material.

### 4.1. Flexural Strength

The presented results showed that the flexural strength of all of the tested materials was sufficient for the single-unit restoration of hard tissue [12,37]. Among the tested materials, the highest flexural strength was demonstrated by GR and CR and the lowest by EN and VSC A. In the case of EN block, despite the high filler content (similar to GR), the recorded strength was the lowest among tested materials. The recorded flexural strength of GR and EN is lower than reported by Lauvahutanon et al. [38] and Ling et al. [39]. The difference can be explained by slightly different sample manufacturing (sample size and surface roughness) and/or variations in the block production process. When analysing the obtained results, the influence of layer angulation during printing on the final flexural strength became visible (VSC A vs. VSC B). The vertical orientation of the layers in relation to the long axis of the sample (VSC A) significantly decreases flexural strength. Recorded value drops from 143.39 (12.88) MPa (VSC A) to 119.85 (17.95) MPa VSC B. However, no traces of delamination of the samples at the boundaries of the layers were observed (SEM), and layer boundaries were also invisible which proves the high homogeneity of the 3D printed material. To summarise, the printed objects should be positioned in such a way that the tensile force generated during mastication is applied along and not across the layers. The relatively low flexural strength of EN, despite high filler content, is a disadvantage of polymer infiltrated ceramic, which was also confirmed by others [25,40]. This may be due to the connection between the ceramic skeleton and the resin matrix. Perhaps it is more difficult to achieve interfacial adhesion during resin infiltration of the sintered porous ceramic matrix. In conclusion, we found no correlation between filler content by volume and flexural strength (R = 0.39), with does not match the results presented by Chung [40] which revealed high correlation (R = 0.89) between filler content by volume and strength. This phenomenon should be related to the different structure of the EN materials.

### 4.2. Flexural Modulus

While flexural strength is the basic parameter that emphasizes the differences between the materials, its role is overestimated. The strength does not provide information about the stiffness of the material and its behaviour during load application. For a better understanding of the strain-stress characteristics of the tested material, calculation of the flexural modulus is been usually done. This parameter is reflected by the slope of the flattened part of the curve during 3-point bending [12,13,39]. During one test, we can calculate both parameters—flexural strength and modulus. The flexural modulus and therefore stiffness of the tested material differ significantly, which results from the differences in their structure and production method. The compared materials are characterised by a large diversity of the flexural modulus. At one end of the scale there is EN, the stiffest of tested materials, with ceramic skeleton responsible for the unique characteristics of this material. EN is significantly stiffer than GR, both with similar filler content. This observation was also confirmed in the studies of other authors [25,39]. At the other end of the scale there is VSC, the material with the lowest stiffness due to the lowest filler content. The relatively low filler content is forced by the need to maintain the liquid consistency necessary in the 3D printing process. The designers of the VSC material have managed to obtain a stable liquid material that does not phase out and the filler does not sink to the bottom of the vat. This facilitates the repeatability of the prints, which was observed by the authors when preparing the bar samples. A low flexural modulus of VSC, significantly lower than that of hard tissues, should be considered a disadvantage [9,11,25,41,42]. Due to the large deflection of the material core under the load, the bonding and the hard tissues may also be locally overloaded [43]. During the analysis of the inorganic filler content measured by volume in the SEM test, Pearson’s linear correlation (R = 0.861) was found between the filler volume and the flexural moduli. This is in line with the results of Mirica et al. (R = 0.89) [44].

### 4.3. Microhardness

The surface hardness of the tested resin composites in this study was assessed using Vickers microhardness test (HV01). This test determines the relative resistance of the material surface to an external force applied by an indentation of cube corner [45]. The microhardness of the tested materials differs significantly and the results are consistent with those reported by Ling et al. [39]. The hardest in this set of materials is EN, which is related to its structure, which includes a ceramic scaffold. GR with a similar filler content has a significantly lower hardness, which may be due to the lack of connections between the filler particles and their smaller dimensions. Despite the differences in the structure of materials, a high positive correlation (R = 0.822) was observed between the filler content (by volume) and the microhardness. A similar relationship was observed by Ling et al. [39], Chung (R = 0.89) [40] and Mirica et al. [44]. The hardness of the material can also be related to the abrasion and materials with low hardness may be more susceptible to wear [46]. It can be therefore assumed that the wear of printed material will progress faster than that of the milled materials. However, this requires confirmation in further studies.

### 4.4. Fractographic and Structure Analysis

The structure of all tested materials is typical for resin composites, where we can see the filler embedded in the resin matrix. However, in the case of EN, bridging between the filler particles and bigger filler particle size is visible. The fracture surface structure of the CR, GR, VSC A and VSC B samples shows crack deflection due to particle bypassing, which is related to the elongation of the fracture path, and thus an increase in the fracture energy expenditure. On the surface of CR and GR samples, large deviations in the crack course are noticeable compared to VSC A and VSC B samples, caused by larger and more diversified particle sizes and higher filler content. The crack after reaching the large filler particle deflects markedly and continues almost perpendicular to the original direction of movement along the interface. The described cracking mechanism significantly increases the fracture toughness of the material. This finding is supported by the recorded flexural strength—significantly higher values of GR and CR flexural strength.

The observed fractures are typical for brittle materials. The direction of crack propagation runs radially from the fracture centre, with visible numerous bending marks perpendicular to the direction of the fracture propagation [14]. The fracture mechanism of EN materials is different because the fracture surface structure of EN specimen shows crack bridging by filler particles. This type of fracture mechanism should increase the energy expenditure of the process and significantly increase the fracture toughness of the material. The filler particles only break when their size in the direction of the crack propagation is clearly smaller than the size of the particles at the point of the crack initiation. The probability of this phenomenon increases with the increase of the size and their quantity in the material. This additional energy expenditure necessary to break the sintered particles of the ceramic skeleton is not reflected in the strength of this material. The strength of EN, unlike microhardness, is significantly lower than that of other milling materials. It can be assumed that the filler particles in GR, CR and VSC B are better connected to the resin matrix and those materials have significantly higher strength.

### 4.5. Fractal and Texture Analysis

The fractal dimension of two dimensioned shapes is focused below the value of 2. The more complex the examined shape, the lower FD. It is confirmed by our study. We observed a very strong linear correlation between the per cent amount of filler in the composite and the value of the fractal dimension. A strong correlation between FD and the mean size of the filler was revealed, too. More filler in a sample, more homogeneity of material observed (higher value of FD revealed). The lowest value of FD was observed in the case of VSC A and VSC B material. The amount of filler and mean size of filler were the lowest of all materials in this composite group. The highest value of FD was seen in the EN material, which was characterized by the highest filler size and 75% of filler in the volume.

It is interesting that FD of ROIs for fracture zone and bending marks on the surface were statistically different in the case of VSC A, VSC B and CR but the difference did not exist between the same ROIs of EN and GR composites.

Salerno et al. applied fractal analysis in the estimation of the roughness of dental restoration after air-polishing [47]. Berezina et al. [48] used fractal dimension analysis of 3D topographic atomic force microscopy images of dental ceramics produced from nanoparticles of alumina and tetragonalzirconia (t-ZrO2) with the addition of Ca ions. Wilson et al. [49] revealed the value of fractal dimension in range 2.19–2.49 in the case of thermoset dimethacrylate polymer nanocomposites. The results of Barszczewska-Rybarek et al. [50] suggest that the fracture behaviour of poly(dimethacrylate) matrix of dental materials can be controlled by fractal morphology. Bulpakdi et al. [51] showed that fractal analysis could be an alternative analytic tool for clinically failed restorations, especially in cases that could not be analysed using other techniques, such as fractography. Gao et al. [52] observed highly positive correlations between the fractal dimension of the fracture surfaces and the impact strength of the cellulose/PLA composites. Multifractal analysis can be used to describe the topography of fracture surfaces [53].

Fracture Origin in VCS A and VCS B were characterised by a fine-grained surface of evenly distributed (high chaos measure values) fine surface irregularities (high ShrtREmp and low LgnREmph). The direction of Crack Propagation ROI has similar textural features on the surface. The lateral bending marks are quite similar in all tested materials.

Significant differences between the materials assessed were shown by the presented results of laboratory tests and image analyses. However, the tests and analyses has some limitations resulting from the lack of simulation of clinical conditions and the ageing process of the materials. It is necessary to evaluate the influence of the ageing process on the resin composite for 3D printing

## 5. Conclusions

Within the limitations of this study, the following conclusions can be drawn:The printed objects should be positioned in such a way that the tensile force generated during mastication is applied along and not across the layers.Flexural strength of VSC depends on layers orientation and is significantly lower than σ*_f_* of GR and CR,Due to the low filler content, flexural modulus of the printed material is the lowest among the tested materials and lower than that of dentin,A strong linear correlation between FD and E*_f_*,; between the filler volume and E*_f_* and between filler volume and HV01 was observed,The texture Index can be recommended to describe the differences in the appearance of the fracture between surfaces.There are no statistical differences between fractal dimension of VSC A and VSC B material.In the VSC A material, more random distribution of inorganic particles is observed in the texture features than in VSC B.

## Figures and Tables

**Figure 1 materials-14-04919-f001:**
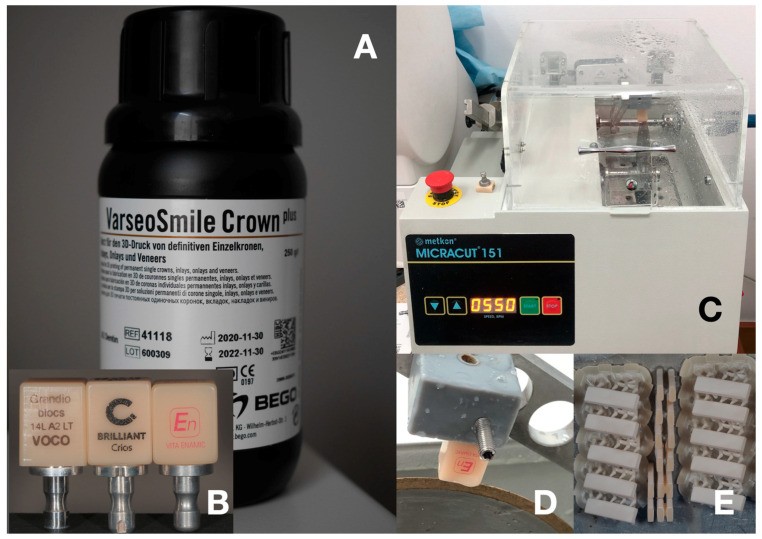
(**A**) bottle of VarseoSmile Crown plus composite resin for 3D printing; (**B**) blocks of materials for milling; (**C**) diamond saw or blocks cutting; (**D**) 3D printed holder with CAD/CAM block during cutting process; (**E**) 3D printed bars of VarseoSmile Crown plus on the printing platform (after cleaning with ethanol).

**Figure 2 materials-14-04919-f002:**
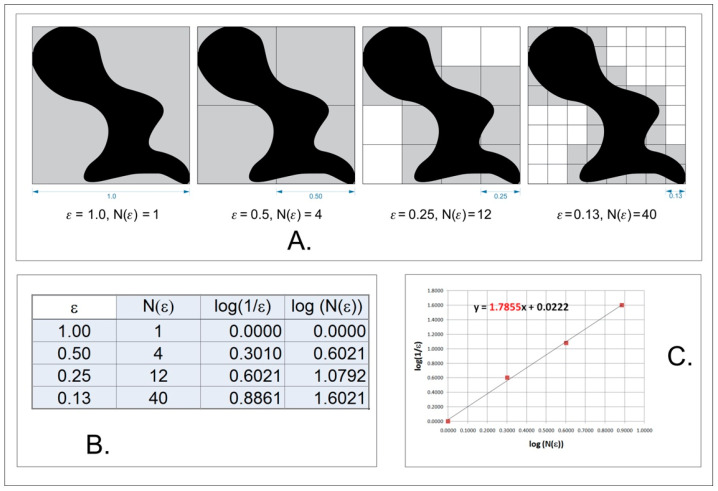
Graphical interpretation of counting box method for fractal dimension counting; (**A**) analysed bitmap, dimension of analysed square size (ε); (**B**) Number of squares need to cover examined shape in the function of square size (ε); (**C**) a straight line drawn through points from table B on the x-y chart in decimal logarithm scale. The slope factor of this straight line is a value fractal dimension counted using the box method.

**Figure 3 materials-14-04919-f003:**
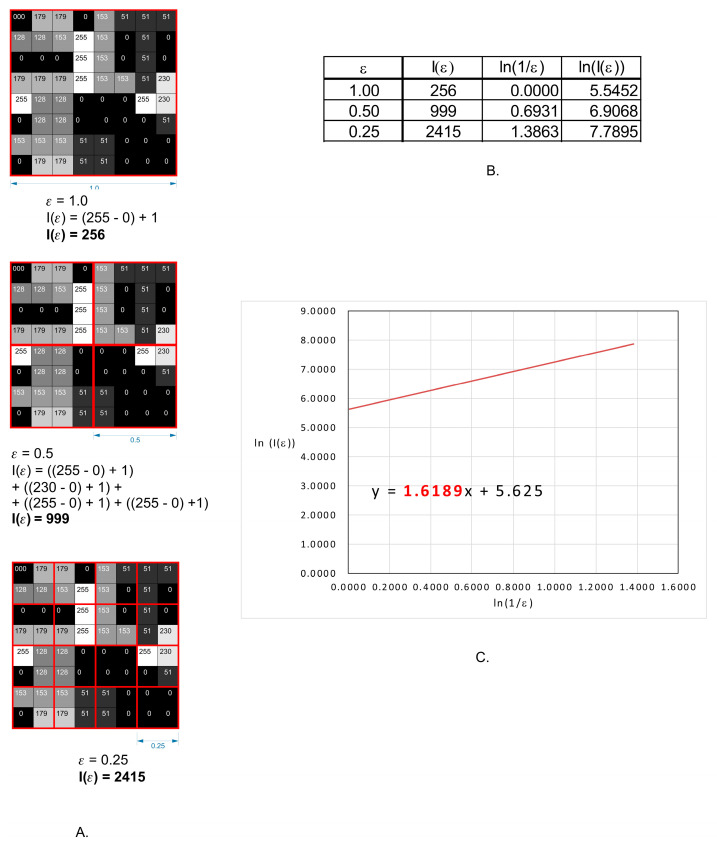
Graphical interpretation of intensity difference algorithm for fractal dimension counting. (**A**) an example of a grayscale 8 bits image, with numbers in squares representing the intensity level of each pixel–0–black, 255 white. Red squares represent scale—ε. (**B**) the values of intensity difference for each step of scale reduction (ε). (**C**) a straight line drawn through points from table B on the x-y chart in the natural logarithm scale. The slope factor for this straight line is a value fractal dimension counted by intense difference algorithm.

**Figure 4 materials-14-04919-f004:**
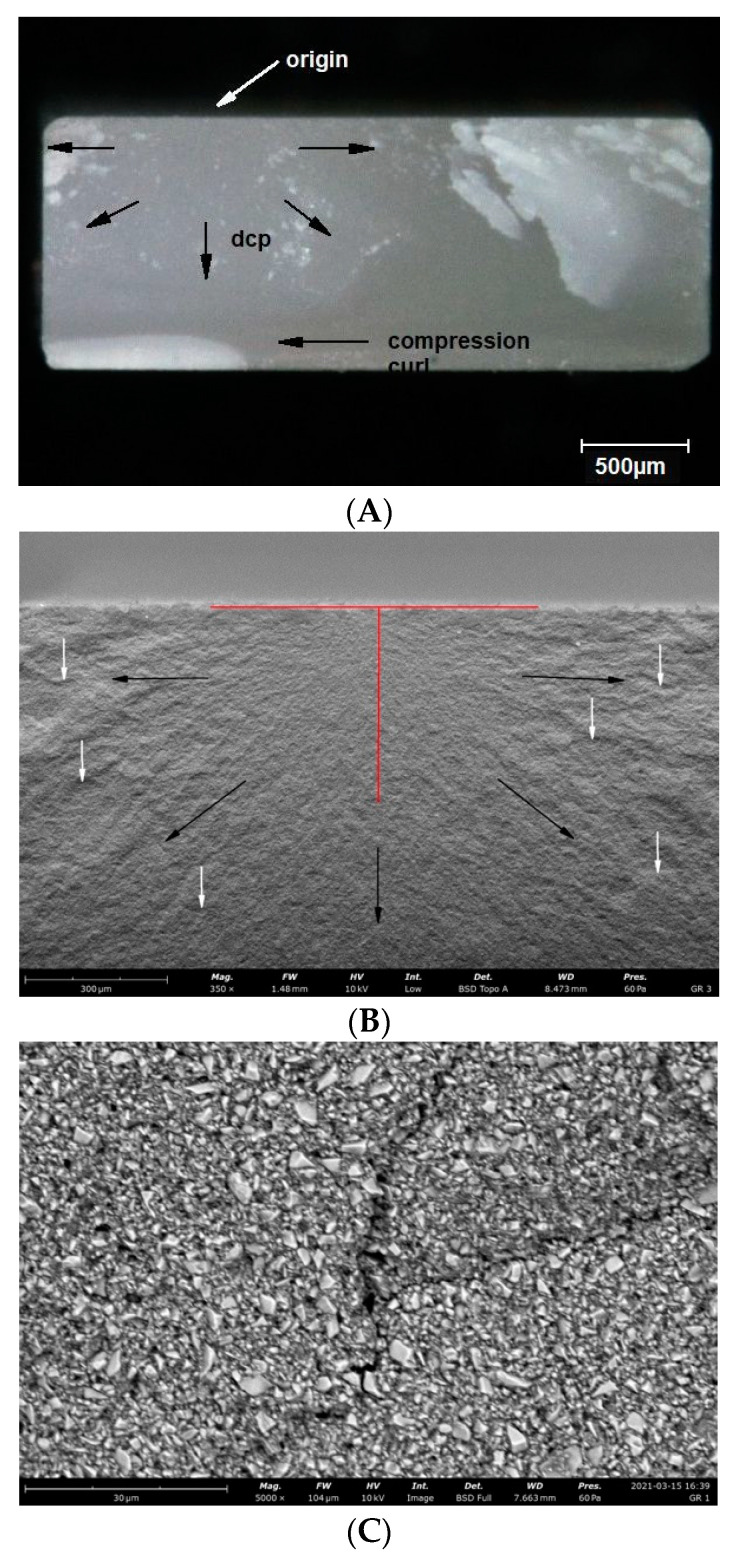
Representative microscopic and SEM images of the fracture surface of the GR sample; (**A**) visible compression curl on top, the direction of crack propagation (DCP) and the origin on the bottom; (**B**) Red lines indicate the width and depth of the crack at the fracture origin. Black arrows indicate the direction of crack propagation away from the crack origin, white arrows indicate bending marks on the fracture surface; (**C**) microcrack spreading along the particle boundaries visible on the enlarged area of figure (**B**) indicated by the white arrow; (**D**) microcrack spreading along the particle boundaries visible on the enlarged area of figure (**B**) indicated by the white arrow.

**Figure 5 materials-14-04919-f005:**
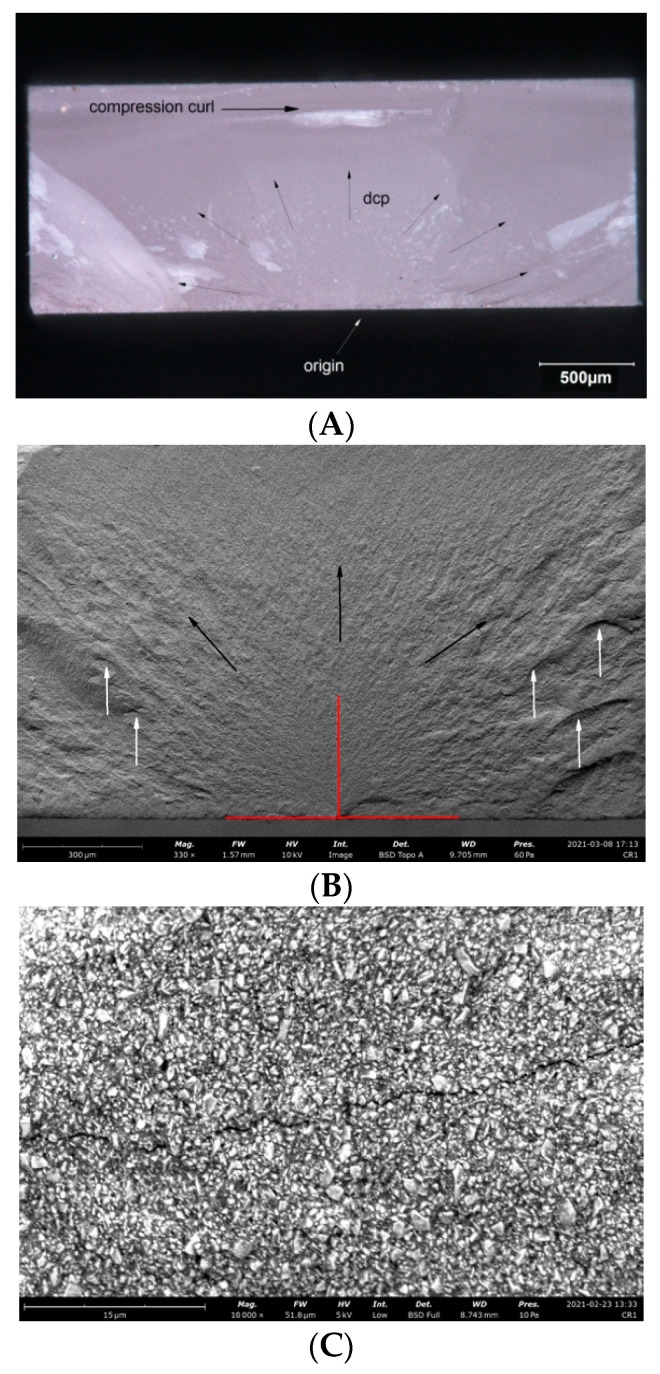
Representative microscopic and SEM images of the fracture surface of the CR sample; (**A**) visible compression curl on top, the direction of crack propagation and the origin on the bottom; (**B**) Red lines indicate the width and depth of the crack at the fracture origin. Black arrows indicate the direction of crack propagation away from the crack origin, white arrows indicate bending marks on the fracture surface; (**C**) microcrack spreading along the particle boundaries visible on the enlarged area of figure (**B**) indicated by the white arrow; (**D**) microcrack spreading along the particle boundaries visible on the enlarged area of figure (**B**) indicated by the white arrow.

**Figure 6 materials-14-04919-f006:**
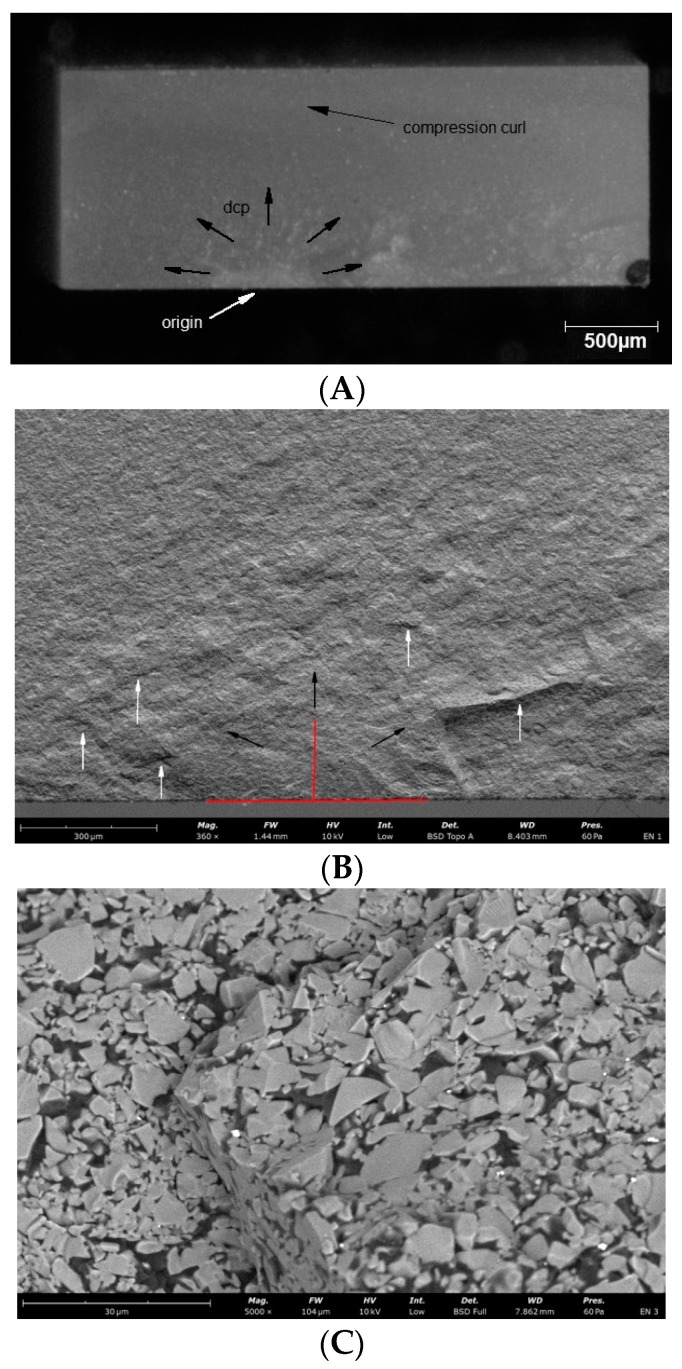
Representative microscopic and SEM images of the fracture surface of the EN sample; (**A**) visible compression curl on top, the direction of crack propagation and the origin on the bottom; (**B**) Red lines indicate the width and depth of the crack at the fracture origin. Black arrows indicate the direction of crack propagation away from the crack origin, white arrows indicate bending marks on the fracture surface; (**C**) microcrack by particles visible on the enlarged area of figure (**B**) indicated by the white arrow; (**D**) microcrack by particles visible on the enlarged area of figure (**B**) indicated by the white arrow.

**Figure 7 materials-14-04919-f007:**
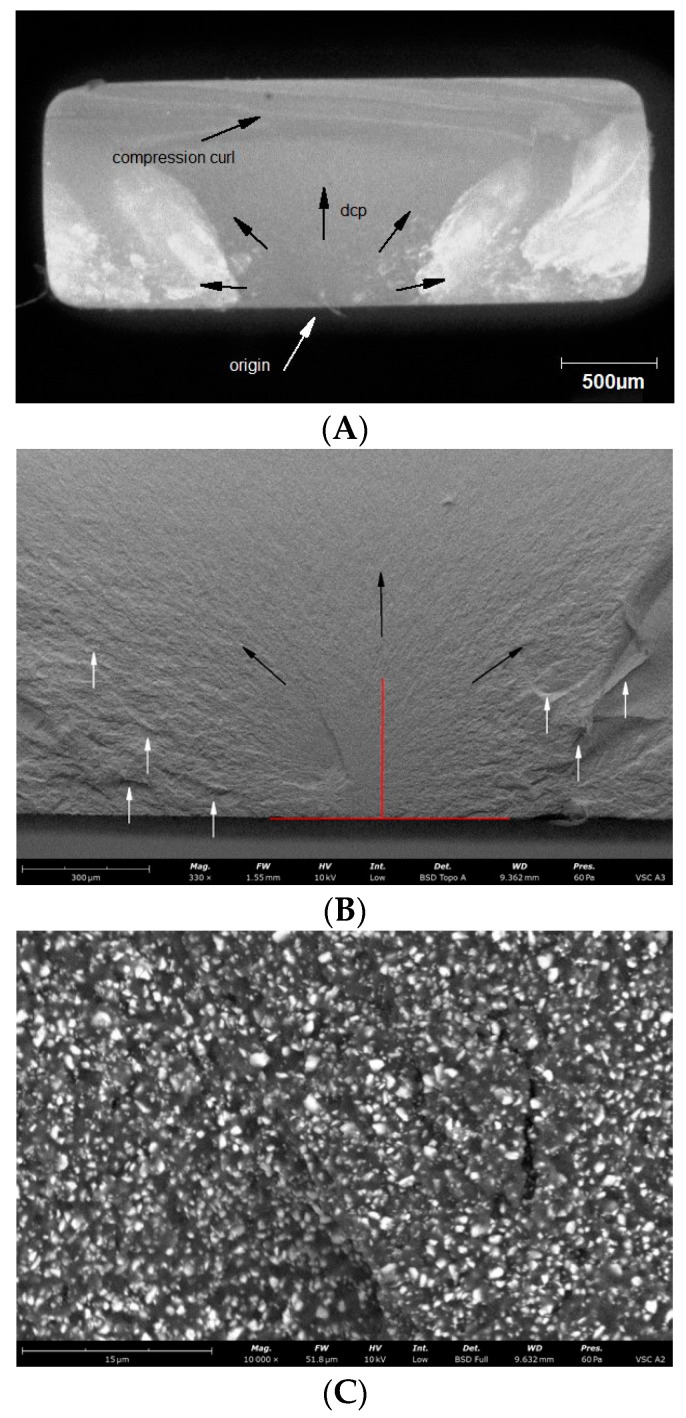
Representative microscopic and SEM images of the fracture surface of the VSC A sample; (**A**) visible compression curl on top, the direction of crack propagation and the origin on the bottom; (**B**) Red lines indicate the width and depth of the crack at the fracture origin. Black arrows indicate the direction of crack propagation away from the crack origin, white arrows indicate bending marks on the fracture surface; (**C**) microcrack spreading along the particle boundaries visible on the enlarged area of figure (**B**) indicated by the white arrow; (**D**) microcrack spreading along the particle boundaries visible on the enlarged area of figure (**B**) indicated by the white arrow.

**Figure 8 materials-14-04919-f008:**
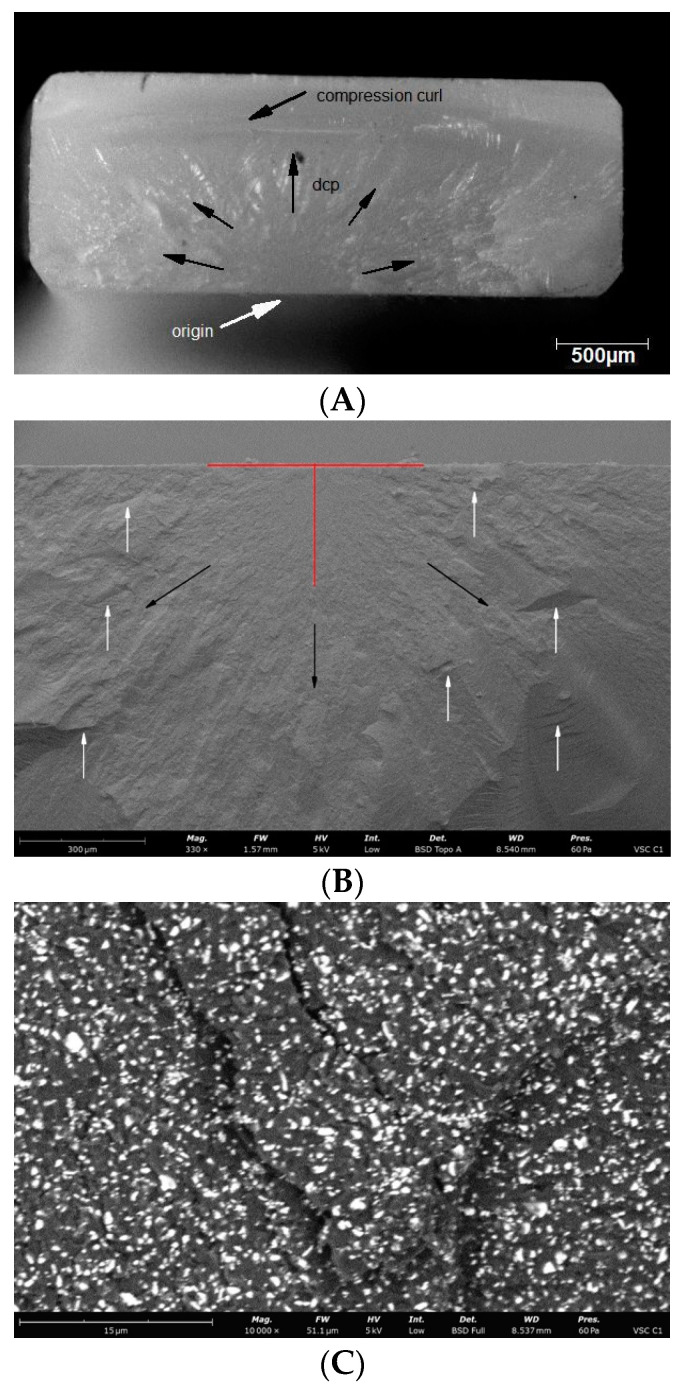
Representative microscopic and SEM images of the fracture surface of the VSC B sample; (**A**) visible compression curl on top, the direction of crack propagation and the origin on the bottom; (**B**) Red lines indicate the width and depth of the crack at the fracture origin. Black arrows indicate the direction of crack propagation away from the crack origin, white arrows indicate bending marks on the fracture surface; (**C**) microcrack spreading along the particle boundaries visible on the enlarged area of figure (**B**) indicated by the white arrow; (**D**) microcrack spreading along the particle boundaries visible on the enlarged area of figure (**B**) indicated by the white arrow.

**Figure 9 materials-14-04919-f009:**
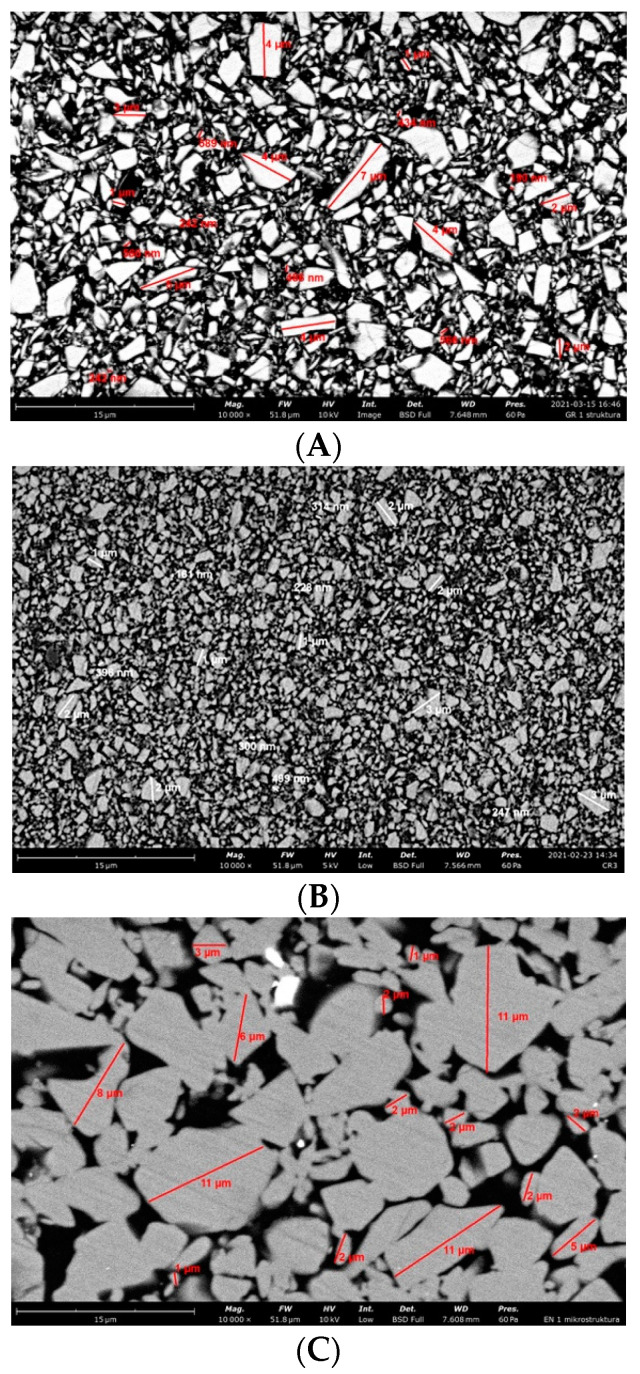
Representative SEM images of the surface of tested materials; (**A**) GR filler size ranged from 190 nm to 7 µm, filler content 70–80 vol. %; (**B**) CR filler size ranged from 160 nm to 3 µm, filler content around 55–65 vol. %; (**C**) EN filler size ranged from 1 µm to 11 µm, filler content around 75 vol. %; (**D**) VSC A filler size ranged 430 nm to 3 µm, filler content around 24–30 vol. %; (**E**) VSC B filler size 430 nm do 3 µm, filler content around 19–24 vol. % (GR—Grandio blocs, CR—Brilliant Crios, EN—Enamic, VSC—Varseo Smile Crown plus).

**Figure 10 materials-14-04919-f010:**
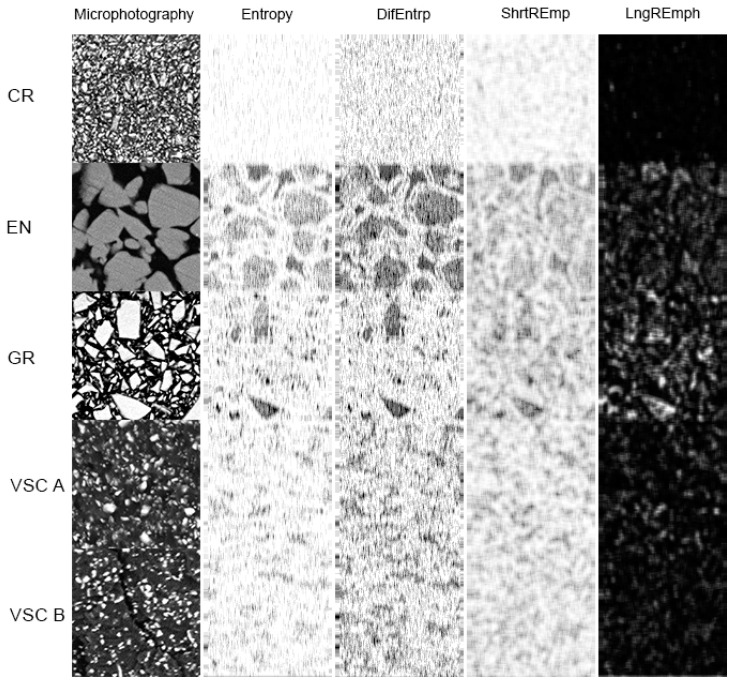
Texture analysis of dental composites samples in two kinds of features in high magnification: ROI = 15 µm × 15 µm. Derived from co-occurrence matrix (Entropy and DifEntrp) and the run-length matrix (ShryREmp and LngREmph). In the feature intensity maps of the polished samples, white indicates a significant intensity of a given texture feature and black indicates none or low intensity of the feature.

**Figure 11 materials-14-04919-f011:**
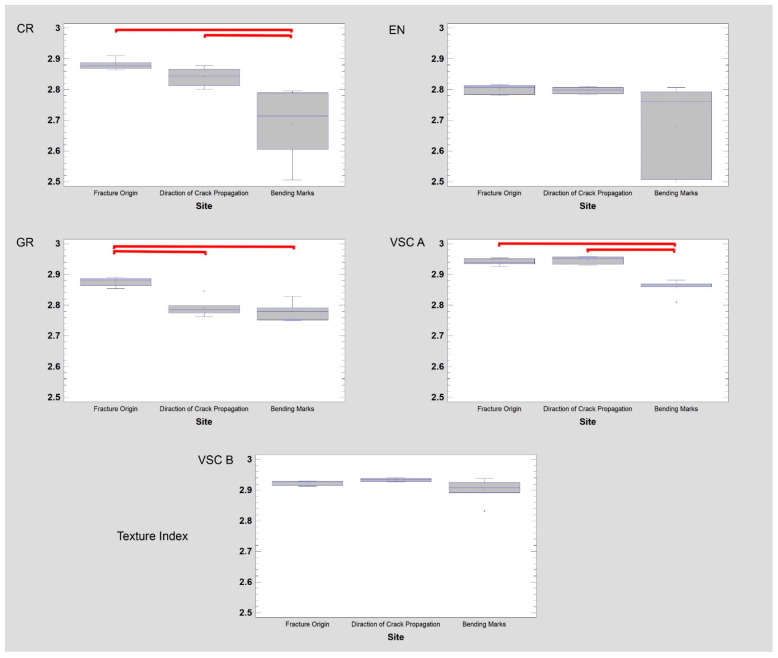
Visible light examination of fracture surfaces—microphotographs of five dental composites. Results of surface texture analysis using Texture Index, GR—Grandio blocs, CR—Brilliant Crios, EN—Enamic, VSC—Varseo Smile Crown plus.

**Table 1 materials-14-04919-t001:** Machinable materials used in the study.

Brand	Abr.	Manufacturer	Composition	Lot No.	Shade	Block Size
Grandio Blocs	GR	VOCO, Cuxhaven, Germany	86 wt % Nanohybride fillers, 14% UDMA + DMA [25,26]	1,711,521	A2 HT	C 14L
Brilliant Crios	CR	Coltene/Whaledent A.G. Altstatten, Switzerland	Resin matrix cross-linked methacrylate, 70.7 wt % barium glass (<1 µm), amorphous silica (<20 nm) [25,27]	H22,667	A2 LT	C 14
Enamic	EN	Vita Zahnfabrik, Bad Sackingen, Germany	14 wt % (25 vol %) methacrylate polymer (UDMA, TEGDMA) and 86 wt % fine-structure feldspar ceramic network [28,29]	56,560	2M2 HT	C 14
VarseoSmile Crown plus	VSC	Bego, Bremen, Germany	4′-isopropylidiphenol, ethoxylated and 2-methylprop-2enoic acid. Silanized dental glass, methyl benzoylfor- mate, diphenyl (2,4,6-trimethylbenzoyl) phosphine oxide, 30–50 wt. %—inorganic fillers (particle size 0.7 μm) [8]	600,309	A2 Dentin	Liquid Resin

UMDA: urethane dimethacrylate; TEGDMA: triethylene glycol dimethacrylate; Bis-GMA: bisphenol A diglycidylether methacrylate; Bis-EMA: ethoxylate bisphenol-A dimethacrylate; DMA: dimethacrylate; Bis-MEEP: 2,2-Bis(4-methacryloxypolyethoxyphenyl) propane; EDMA—ethyleneglycoldimethacrylate; DMA—dimethacrylate.

**Table 2 materials-14-04919-t002:** Mechanical properties of the testing materials.

Material	σ*_f_* [MPa]	E*_f_* [GPa]	HV01	Filler	Filler
	$\bar{x}\;(\mathrm{SD})$	$\bar{x}\;(\mathrm{SD})$	$\bar{x}\;(\mathrm{SD})$	Vol. %	Size
**GR**	186.02 (10.49) *	16.95 (0.50) *	140.43 (5.47) *	70–80	190 nm to 7 µm
**CR**	170.29 (9.41) *	11.14 (0.17) *	75.40 (2.18) *	55–65	160 nm to 3 µm
**EN**	118.96 (2.81) A	28.55 (0.34)	273.42 (27.11)	75	1 µm to 11 µm
**VSC A**	119.85 (17.95) A	4.37 (0.8) B	25.8 (0.7) C	24–30	430 nm to 3 µm
**VSC B**	143.39 (12.88)	4.69 (0.15) B	28.16 (1.42) C	19–24	430 nm to 2 µm
*p* value *	*p* < 0.001	*p* < 0.001	*p* < 0.001		

Where: σ*_f_*—flexural strength; E*_f_*—flexural modulus; HV01—Vickers microhardness; Materials with the same letter within a column are not significantly different (*p* > 0.05); Mean values (*n* = 10) and standard deviations in parentheses; $\bar{x}$—mean; SD—standard deviation; *—data previously published by the authors [10]; GR—Grandio blocs, CR—Brilliant Crios, EN—Enamic, VSC—Varseo Smile Crown plus.

**Table 3 materials-14-04919-t003:** Mean values of FD for ROI 15 µm × 15 µm (FD—fractal dimension, SD—standard deviation).

Mean Values of Fractal Dimension for ROI 15 μm × 15 μm					
Material	VSC A	VSC B	CR	EN	GR
Mean FD	1.541	1.550	1.791	1.899	1.769
SD	0.044	0.025	0.033	0.013	0.009

**Table 4 materials-14-04919-t004:** Results of post-hoc (least significant difference) ANOVA fractal dimension value between the microstructure of all materials (ROI—15 µm × 15 µm), underlined—*p* < 0.05, significant statistical difference (FD—fractal dimension).

vs.	CR	EN	GR	VSC A	VSC B
CR		0.0000	0.1846	0.0000	0.0000
EN	0.0000		0.0000	0.0000	0.0000
GR	0.1846	0.0000		0.0000	0.0000
VSC A	0.0000	0.0000	0.0000		0.6527
VSC B	0.0000	0.0000	0.0000	0.6527	

**Table 5 materials-14-04919-t005:** Mean values of fractal dimension of fracture zones (ROI 100 µm × 100 µm), FD—fractal dimension, SD—standard deviation.

Mean Values of Fractal Dimension of Fracture Zones (ROI 100 µm × 100 µm)			
Fracture Zone	Fracture Origin	Direction of Crack Propagation	Bending Marks on the Fracture Surface
VSC A			
Mean	1.770	1.773	1.724
SD	0.023	0.008	0.023
VSC B			
Mean	1.780	1.759	1.735
SD	0.020	0.015	0.018
CR			
Mean	1.767	1.740	1.693
SD	0.007	0.013	0.019
EN			
Mean	1.702	1.706	1.687
SD	0.015	0.011	0.021
GR			
Mean	1.742	1.717	1.715
SD	0.016	0.026	0.019

**Table 6 materials-14-04919-t006:** The results of post-hoc (least significant difference) ANOVA between FD value of the same fracture zone of all examined materials for ROI size 100 µm × 100 µm, underlined—*p* < 0.05, significant statistical difference, FD—fractal dimension.

**FD of Fracture Origin**					
**vs.**	**CR**	**EN**	**GR**	**VSC A**	**VSC B**
**CR**		0.000001	0.017799	0.735889	0.175320
**EN**	0.000001		0.000435	0.000000	0.000000
**GR**	0.017799	0.000435		0.008078	0.000590
**VSC A**	0.735889	0.000000	0.008078		0.302052
**VSC B**	0.175320	0.000000	0.000590	0.302052	
**FD of Direction of Crack Propagation**					
**vs.**	**CR**	**EN**	**GR**	**VSC A**	**VSC B**
**CR**		0.000967	0.017239	0.001249	0.045387
**EN**	0.000967		0.246263	0.000000	0.000004
**GR**	0.017239	0.246263		0.000002	0.000091
**VSC A**	0.001249	0.000000	0.000002		0.138324
**VSC B**	0.045387	0.000004	0.000091	0.138324	
**FD of Bending Marks on the Fracture Surface**					
**vs.**	**CR**	**EN**	**GR**	**VSC A**	**VSC B**
**CR**		0.573833	0.073630	0.013429	0.001231
**EN**	0.573833		0.022256	0.003449	0.000286
**GR**	0.073630	0.022256		0.435122	0.087909
**VSC A**	0.013429	0.003449	0.435122		0.335130
**VSC B**	0.001231	0.000286	0.087909	0.335130	

**Table 7 materials-14-04919-t007:** The results of post-hoc (least significant difference) ANOVA between fractal dimension of fracture origin (Origin), direction of crack propagation (DCP) and bending marks on the fracture surface (BM) inside the same material (ROI—100 µm × 100 µm), underlined font—significant statistical difference, n.s.—no significant differences *p* > 0.05 in ANOVA.

**VSC A**			
**vs.**	**Origin**	**DCP**	**BM**
**Origin**		0.785391	0.000755
**DCP**	0.785391		0.000433
**BM**	0.000755	0.000433	
**VSC B**			
**vs.**	**Origin**	**DCP**	**BM**
**Origin**		0.056383	0.000484
**DCP**	0.056383		0.031915
**BM**	0.000484	0.031915	
**CR**			
**vs.**	**Origin**	**DCP**	**BM**
**Origin**		0.004370	0.000000
**DCP**	0.004370		0.000028
**BM**	0.000000	0.000028	
**EN**			
**vs.**	**Origin**	**DCP**	**BM**
**Origin**	n.s.	n.s.	n.s.
**DCP**	n.s.	n.s.	n.s.
**BM**	n.s.	n.s.	n.s.
**GR**			
**vs.**	**Origin**	**DCP**	**BM**
**Origin**	n.s.	n.s.	n.s.
**DCP**	n.s.	n.s.	n.s.
**BM**	n.s.	n.s.	n.s.

**Table 8 materials-14-04919-t008:** Comparison of texture characteristics of tested resin composites. The numerical expression of the variability given in Figure 10 (DifEntrp—difference entropy, ShrtREmph—short run-length emphasis moment, LngREmph—long run-length emphasis moment).

Material	Entropy	DifEntrp	ShrtREmp	LngREmph	Texture Index	Composite Index
CR	3.09 ± 0.03	1.94 ± 0.01	0.96 ± 0.00	1.20 ± 0.04	2.57 ± 0.09	1.55 ± 0.00
EN	2.61 ± 0.12	1.29 ± 0.04	0.89 ± 0.02	1.66 ± 0.15	1.58 ± 0.18	1.45 ± 0.04
GR	2.39 ± 0.03	1.37 ± 0.01	0.85 ± 0.00	3.20 ± 0.69	0.75 ± 0.08	1.61 ± 0.01
VSC A	2.86 ± 0.02	1.45 ± 0.01	0.91 ± 0.00	1.54 ± 0.02	1.86 ± 0.03	1.59 ± 0.01
VSC B	2.77 ± 0.02	1.42 ± 0.01	0.91 ± 0.00	1.51 ± 0.04	1.83 ± 0.06	1.56 ± 0.01

**Table 9 materials-14-04919-t009:** The information in Table 8 supplemented with data on the statistical significance of the differences detected. The Least Significant Difference procedure (indicates the magnitude of the limits indicating the smallest difference between any two means that can be declared to represent a statistically significant difference) DifEntrp—difference entropy, ShrtREmph—short run-length emphasis moment, LngREmph—long run-length emphasis moment.

Contrast	Entropy	DifEntrp	ShrtREmph	LngREmph	Texture Index	Composite Index
	Difference	Difference	Difference	Difference	Difference	Difference
CR–EN	0.48 *	0.20 *	0.07 *	−0.46 *	0.99 *	0.10 *
CR–GR	0.70 *	0.12 *	0.11 *	−1.99 *	1.82 *	−0.06 *
CR–VSC A	0.23 *	0.04 *	0.05 *	−0.34 *	0.72 *	−0.04 *
CR–VSC B	0.32 *	0.07 *	0.05 *	−0.31 *	0.74 *	−0.01 *
EN–GR	0.22 *	−0.08 *	0.04 *	−1.53 *	0.83 *	−0.15 *
EN–VSC A	−0.25 *	−0.16 *	−0.02 *	0.12	−0.27 *	−0.14 *
EN–VSC B	−0.15 *	−0.13 *	−0.02 *	0.15	−0.25 *	−0.11 *
GR–VSC A	−0.47 *	−0.08 *	−0.06 *	1.66 *	−1.10 *	0.02 *
GR–VSC B	−0.38 *	−0.05 *	−0.06 *	1.68 *	−1.08 *	0.04 *
VSC A–VSC B	0.10 *	0.03 *	−0.00	0.03	0.03	0.03 *

* denotes a statistically significant difference at *p* < 0.05.

**Table 10 materials-14-04919-t010:** Texture features of the origin site of material fracture, DifEntrp—difference entropy, ShrtREmph—short run-length emphasis moment, LngREmph—long run-length emphasis moment.

Material	Entropy	DifEntrp	ShrtREmp	LngREmph	Texture Index	Composite Index
CR	3.21 ± 0.01	1.49 ± 0.00	0.97 ± 0.00	1.11 ± 0.00	2.88 ± 0.02	1.53 ± 0.00
EN	3.20 ± 0.01	1.44 ± 0.01	0.97 ± 0.00	1.14 ± 0.00	2.80 ± 0.02	1.49 ± 0.01
GR	3.23 ± 0.01	1.48 ± 0.01	0.97 ± 0.00	1.12 ± 0.00	2.88 ± 0.01	1.53 ± 0.00
VSC A	3.22 ± 0.00	1.49 ± 0.00	0.98 ± 0.00	1.10 ± 0.00	2.94 ± 0.01	1.52 ± 0.01
VSC B	3.20 ± 0.00	1.49 ± 0.00	0.98 ± 0.00	1.09 ± 0.00	2.92 ± 0.01	1.52 ± 0.00

**Table 11 materials-14-04919-t011:** The information in Table 10 supplemented with data on the statistical significance of the differences detected in the site of fracture origin. The Least Significant Difference procedure (indicates the magnitude of the limits indicating the smallest difference between any two means that can be declared to represent a statistically significant difference), DifEntrp—difference entropy, ShrtREmph—short run-length emphasis moment, LngREmph—long run-length emphasis moment.

Contrast	Entropy	DifEntrp	ShrtREmp	LngREmph	Texture Index	Composite Index
	Difference	Difference	Difference	Difference	Difference	Difference
CR–EN	0.01	0.04 *	0.01 *	−0.03	0.08 *	0.04 *
CR–GR	−0.02 *	0.01	0.00 *	−0.01	0.00	0.00
CR–VSC A	−0.01 *	−0.00	−0.00 *	0.02 *	−0.06 *	0.01
CR–VSC B	0.01	0.00	−0.00 *	0.02 *	−0.04 *	0.01
EN–GR	−0.02 *	−0.04 *	−0.00 *	0.02 *	−0.07 *	−0.03 *
EN–VSC A	−0.02 *	−0.05	−0.01 *	0.05 *	−0.14 *	−0.03 *
EN–VSC B	0.01	−0.04 *	−0.01 *	0.05 *	−0.12 *	−0.03 *
GR–VSC A	0.01	−0.01	−0.01 *	0.03 *	−0.06 *	0.00
GR–VSC B	0.03 *	−0.00	−0.01 *	0.03 *	−0.05 *	0.01
VSC A–VSC B	0.02 *	0.00	−0.00	0.00	0.02 *	0.00

* denotes a statistically significant difference at *p* < 0.05.

**Table 12 materials-14-04919-t012:** Texture features of the fracture propagation area of materials investigated, DifEntrp—difference entropy, ShrtREmph—short run-length emphasis moment, LngREmph—long run-length emphasis moment.

Material	Entropy	DifEntrp	ShrtREmp	LngREmph	Texture Index	Composite Index
CR	3.21 ± 0.00	1.46 ± 0.02	0.97 ± 0.00	1.13 ± 0.01	2.84 ± 0.03	1.51 ± 0.02
EN	3.21 ± 0.00	1.44 ± 0.01	0.97 ± 0.00	1.15 ± 0.00	2.80 ± 0.01	1.49 ± 0.01
GR	3.21 ± 0.01	1.45 ± 0.02	0.97 ± 0.00	1.15 ± 0.01	2.79 ± 0.03	1.50 ± 0.02
VSC A	3.23 ± 0.00	1.48 ± 0.00	0.98 ± 0.00	1.10 ± 0.00	2.95 ± 0.01	1.52 ± 0.00
VSC B	3.22 ± 0.01	1.47 ± 0.01	0.98 ± 0.00	1.10 ± 0.00	2.93 ± 0.01	1.51 ± 0.01

**Table 13 materials-14-04919-t013:** The information in Table 12 supplemented with data on the statistical significance of the differences detected in the fracture propagation area. The Least Significant Difference procedure (indicates the magnitude of the limits indicating the smallest difference between any two means that can be declared to represent a statistically significant difference), DifEntrp—difference entropy, ShrtREmph—short run-length emphasis moment, LngREmph—long run-length emphasis moment.

Contrast	Entropy	DifEntrp	ShrtREmp	LngREmph	Texture Index	Composite Index
	Difference	Difference	Difference	Difference	Difference	Difference
CR–EN	0.00	0.02 *	0.00 *	−0.02 *	0.04 *	0.02 *
CR–GR	0.00	0.02	0.00 *	−0.02 *	0.05 *	0.01
CR–VSC A	−0.02 *	−0.02 *	−0.01 *	0.03 *	−0.11 *	−0.01
CR–VSC B	−0.01	−0.01	−0.01 *	0.03 *	−0.09	0.00
EN–GR	−0.00	−0.01	0.00	0.00	0.01	−0.01
EN–VSC A	−0.02 *	−0.05 *	−0.01 *	0.05 *	−0.15 *	−0.03 *
EN–VSC B	−0.01 *	−0.03 *	−0.01 *	0.05 *	−0.14 *	−0.02 *
GR–VSC A	−0.02 *	−0.04 *	−0.01 *	0.05 *	−0.16 *	−0.02 *
GR–VSC B	−0.01 *	−0.03 *	−0.01 *	0.05 *	−0.14 *	−0.01
VSC A–VSC B	0.01 *	0.01	0.00	0.00	0.01	0.01

* denotes a statistically significant difference at *p* < 0.05.

**Table 14 materials-14-04919-t014:** Texture features of the bending marks of fracture surfaces investigated, DifEntrp—difference entropy, ShrtREmph—short run-length emphasis moment, LngREmph—long run-length emphasis moment.

Material	Entropy	DifEntrp	ShrtREmp	LngREmph	Texture Index	Composite Index
CR	3.17 ± 0.04	1.37 ± 0.05	0.96 ± 0.01	1.18 ± 0.04	2.69 ± 0.11	1.43 ± 0.04
EN	3.17 ± 0.06	1.37 ± 0.08	0.96 ± 0.01	1.19 ± 0.05	2.68 ± 0.16	1.43 ± 0.07
GR	3.21 ± 0.01	1.43 ± 0.01	0.96 ± 0.00	1.15 ± 0.01	2.78 ± 0.03	1.48 ± 0.01
VSC A	3.20 ± 0.01	1.44 ± 0.03	0.97 ± 0.00	1.12 ± 0.01	2.86 ± 0.03	1.48 ± 0.03
VSC B	3.21 ± 0.01	1.44 ± 0.03	0.97 ± 0.00	1.11 ± 0.01	2.90 ± 0.04	1.48 ± 0.03

**Table 15 materials-14-04919-t015:** The information in Table 12 supplemented with data on the statistical significance of the differences detected in the bending marks of fracture surfaces. The Least Significant Difference procedure (indicates the magnitude of the limits indicating the smallest difference between any two means that can be declared to represent a statistically significant difference), DifEntrp—difference entropy, ShrtREmph—short run-length emphasis moment, LngREmph—long run-length emphasis moment.

Contrast	Entropy	DifEntrp	ShrtREmp	LngREmph	Texture Index	Composite Index
	Difference	Difference	Difference	Difference	Difference	Difference
CR–EN	0.00	0.00	0.00	0.00	0.01	0.00
CR–GR	−0.03	−0.05	0.00	0.03	−0.09	−0.05
CR–VSC A	−0.03	−0.06 *	−0.01 *	0.06 *	−0.17 *	−0.05
CR–VSC B	−0.04	−0.07 *	−0.01 *	0.07 *	−0.21 *	−0.05
EN–GR	−0.04	−0.05	−0.01 *	0.03	−0.10	−0.05
EN–VSC A	−0.03	−0.06 *	−0.01 *	0.07 *	−0.18 *	−0.04
EN–VSC B	−0.04	−0.07 *	−0.02 *	0.08 *	−0.22 *	−0.05
GR–VSC A	0.00	−0.01	−0.01 *	0.03	−0.08	0.00
GR–VSC B	−0.01	−0.01	−0.01 *	0.05 *	−0.12 *	0.00
VSC A–VSC B	−0.01	−0.01	0.00	0.01	−0.04	−0.00

* denotes a statistically significant difference at *p* < 0.05.

## Data Availability

Data available from the authors: Wojciech.grzebieluch@umed.wroc.pl; kjurczysz@interia.pl.

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
