# Peer review of "Printable and Machinable Dental Restorative Composites for CAD/CAM Application—Comparison of Mechanical Properties, Fractographic, Texture and Fractal Dimension Analysis"

_materials, 2021, doi:10.3390/ma14174919_

Round 1
Reviewer 1 Report
General
#1.The rationale for undertaking the experiments is not clear and thus, the importance of the result are not stated. This work is materials testing rather than research. The authors should put some more effort in revising and reading through the manuscript before submission.
Abstract.
#2.The purpose of the investigation, comparing properties of milled and 3Dprinted test specimens is not stated in the abstract and only revealed in the M&M section.
#3. The results of the fractographic, texture and fractal dimension analysis are not mentioned
#4. Last sentence unclear
Introduction
#5. The introduction is not addressing the problem(s) of the study.
#6. A number of statements are without citations
#7. The author describe how fracture analyses and Texture analysis are performed. This is part of M&M. The introduction should explain to the reader the implications from fracture analysis and texture analysis.
#8. There is no hypothesis (see general comment)
Materials and method
#9. Many paragraphs need to shortened in order to be more precise and to the point. It is unclear how the filler content is measured
Results
#10. First paragraph and Page 14, last paragraph are instruction to authors, not part of the manuscript
#11. Result section must be condensed and double documentation (Table and text) avoided to make it more readable.
#13. Tables 3- 15: All tables should be self-explanatory. The authors must check the table legends to secure that abbreviation, units etc are correct and included.
Discussion
#14. Must be shortened and limited to a discussion of the results and the interpretation of the results.
#15. Results should not be repeated, unless they are key results.
#16. The limitations of the study is not described.
Conclusion
#17. First paragraph is instruction to authors, not part of the manuscript
#18. It is concluded that “Printed restorations can be as durable as milled restorations”. As far as the reviewer can see, duration of restorations is not investigated. Please delete.
#19. New conclusions is recommended based on a stated rationale for the experiments.
Reviewer 2 Report
General comments
This study examined the various mechanical properties of three kinds of machinable resin composite blocks and one printable resin composite, and analyzed their fractography, texture and fractal dimension after the three-point bending test. The mechanical properties of those materials have been already reported in the previous studies; however, the results of fractography, texture and fractal dimension analysis after the three-point bending test are interesting.
Some questionable and insufficient issues were found in this paper.
Abstract
Line 24
The word of “CNC” is abbreviation. The authors should mention it as “Computerized numerical control (CNC)”.
Line 30, 33 and 38
Each number of (2), (3) and (4) for each sentence should be eliminated.
Introduction
Line 109-112
The description of the aim of this study is unclear. A mere comparison of various properties among the four commercially available materials is unsuitable for the aim of a research. Moreover, the null hypothesis is not mentioned. The focus of this study might be to clarify the various mechanical properties of printable resin composite in comparison to those of machinable resin composite block. Please change the description of the aim of this study more clearly, and mention the null hypothesis.
Materials and Methods section
Line 155-177
The explain of mechanical testing is long. Please make it more concise. The calculation formula for flexural strength and flexural modulus is well-known; therefore, they are able to be eliminated.
Line 192-255
The explain of fractal dimension analysis is too long. Please make it more concise.
Line 204
The description of “Error! Reference source not found.” should be eliminated, or the sentence including the description should be eliminated.
Line 258-280
I suppose that the expression of the calculation formula for Texture and Composite Index may be unnecessary.
Results section
Table 4, 6, 7, 9, 11, 13 and 15 are not necessary. Instead, p-value should be shown at the end of the sentence which the result of the statistical analysis is explained.
Discussion section
Line 602, 611 and 687
The reference numbers shown in the parenthesis are superscript. Please change them.
Line 635
Please eliminate the underline of “done”.
Conclusion
The first sentence of “This section is not mandatory but can be added to the manuscript if the discussion is unusually long or complex.” should be eliminated.
Reviewer 3 Report
You can find attached the PDF file.

Reviewer 4 Report
dear authors, the use of adhesive materials together with CAD-CAM technologies represents today one of the most interesting topics of research in adhesive dentistry. for this reason, the study of the mechanical characteristics of composites obtained by printing or milling can add interesting information both from a scientific and clinical point of view.
some improvements could be made before publication.
abstracts.
the abstract is well presented. however, the aim of the work should be better clarified. at the moment only the tests carried out but not the real purpose of the work are apparent.
has a statistical analysis been done? should be included in the abstract.
the conclusion is not clear. please reword after improving the presentation of the results and inserting the statistical analysis. there are also numbers between parantheses. the meaning is not clear.
introduction
the introduction presents the techniques of producing restorative materials. however, since this is a study where mechanical properties are investigated, these restorative concepts could be introduced in the introduction. what is the state of the art? what materials are recognized as the gold standard? is there evidence of this?
Here is an example of the most recent works in the literature
(Haddadi Y, Ranjkesh B, Isidor F, Bahrami G. Marginal and internal fit of crowns based on additive or subtractive manufacturing. Biomater Investig Dent. 2021 Jun 26;8(1):87-91. doi: 10.1080/26415275.2021.1938576. PMID: 34240060; PMCID: PMC8238058.)
it is right that the tests performed are introduced but how specifically they are evaluated is pertinent to the materials and methods. please move this information.
the aim of the paper should be improved and a null hypothesis added. what should the reader expect from this study? could the materials have different characteristics?
the references do not always follow the statements made. some parts seem to be considerations of the authors. please add references where needed.
for example fractal analysis is not supported by any citation. what are the current fields of use? for example in the study of oral carcinomas. please look at the reference
(D'Addazio G, Artese L, Traini T, Rubini C, Caputi S, Sinjari B. Immunohistochemical study of osteopontin in oral squamous cell carcinoma allied to fractal dimension. J Biol Regul Homeost Agents. 2018 Jul-Aug;32(4):1033-1038. PMID: 30043590).
Materials and Methods.
the study design is interesting. however, it could be presented in a more schematic way so that the reader immediately understands which groups are being examined and how the samples were obtained.
line 204. the reference to the image is missing
it would be interesting to show images of the preparation of the samples to facilitate the repeatability of the study
the results are complete and well presented. the statistical analysis well reveals the differences between the groups
the discussion is well written and the right comparisons are presented with respect to the works in the literature. a correlation (R) is also presented that is not included in the statistical part of the materials and methods. can this part be improved?
some errors with respect to the journal layout are present. we recommend a careful review of the entire manuscript paying attention to the journal guidelines. also the references should be checked.
Round 2
Reviewer 1 Report
#1. Line 118: The null hypothesis was that there would be significant differences in flexural strength,…..
Definition of null hypothesis: A null hypothesis proposes that there is no difference between certain characteristics of a population or data-generating process.
Please correct.
#2. Line 126: microstructure rather than microsrticture
#3. Line 654: Please correct statement to follow the definition of a null hypothesis.
#4. In the first review, it was suggested that the discussion must be shortened and limited to a discussion of the results and the interpretation of the results.
I cannot see that this has been done, and a revision is strongly recommended.
Reviewer 2 Report
This paper was well revised according to reviewer's comments.
